# Learning the Step-size Policy for the Limited-Memory Broyden-Fletcher-Goldfarb-Shanno Algorithm

## Abstract

We consider the problem of how to learn a step-size policy for the Limited-Memory Broyden-Fletcher-Goldfarb-Shanno (L-BFGS) algorithm. This is a limited computational memory quasi-Newton method widely used for deterministic unconstrained optimization but currently avoided in large-scale problems for requiring step sizes to be provided at each iteration. Existing methodologies for the step size selection for L-BFGS use heuristic tuning of design parameters and massive re-evaluations of the objective function and gradient to find appropriate step-lengths. We propose a neural network architecture with local information of the current iterate as the input. The step-length policy is learned from data of similar optimization problems, avoids additional evaluations of the objective function, and guarantees that the output step remains inside a pre-defined interval. The corresponding training procedure is formulated as a stochastic optimization problem using the backpropagation through time algorithm. The performance of the proposed method is evaluated on the training of classifiers for the MNIST database for handwritten digits and for CIFAR-10. The results show that the proposed algorithm outperforms heuristically tuned optimizers such as ADAM, RMSprop, L-BFGS with a backtracking line search and L-BFGS with a constant step size. The numerical results also show that a learned policy can be used as a warm-start to train new policies for different problems after a few additional training steps, highlighting its potential use in multiple large-scale optimization problems.

## 1 Introduction

Consider the unconstrained optimization problem

$$\underset{\boldsymbol{x}}{\text{minimize}} \; f(\boldsymbol{x}) \tag{1}$$

where $f : \mathbb{R}^n \to \mathbb{R}$ is an objective function that is differentiable for all $\boldsymbol{x} \in \mathbb{R}^n$, with $n$ being the number of decision variables forming $\boldsymbol{x}$. Let $\nabla_{\boldsymbol{x}} f(\boldsymbol{x}_0)$ be the gradient of $f(\boldsymbol{x})$ evaluated at some $\boldsymbol{x}_0 \in \mathbb{R}^n$. A general quasi-Newton algorithm for solving this problem iterates

$$\boldsymbol{x}_{k+1} = \boldsymbol{x}_k - t_k \boldsymbol{H}_k \boldsymbol{g}_k \tag{2}$$

for an initial $\boldsymbol{x}_0 \in \mathbb{R}^n$ until a given stop criterion is met. At the $k$-th iteration, $\boldsymbol{g}_k = \nabla_{\boldsymbol{x}} f(\boldsymbol{x}_k)$ is the gradient, $\boldsymbol{H}_k$ is a positive-definite matrix satisfying the secant equation (Nocedal and Wright, 2006, p. 137) and $t_k$ is the step size.

In this paper, we develop a policy that learns to suitably determine step sizes $t_k$ when the product $\boldsymbol{H}_k \boldsymbol{g}_k$ is calculated by the Limited-Memory Broyden–Fletcher–Goldfarb–Shanno (L-BFGS) algorithm (Liu and Nocedal, 1989). The main contributions of the paper are:

1. We propose a neural network architecture defining this policy taking as input local information of the current iterate. In contrast with more standard strategies, this policy is tuning-free and avoids re-evaluations of the objective function and gradients at each step. The training procedure is formulated as a stochastic optimization problem and can be performed by easily applying backpropagation through time (TBPTT).

2. Training classifiers in the MNIST database (LeCun et al., 1998), our approach is competitive against heuristically tuned optimization procedures. Our tests show that the proposed policy is not only able to outperform competitors such as ADAM and RMSprop in wall-clock time and optimal/final value, but also performs better than L-BFGS with backtracking line searches, which is the gold standard, and with constant step sizes, which is the baseline.

3. According to subsequent experiments on CIFAR-10 (Krizhevsky et al., 2009), the proposed policy can generalize to different classes of problems after a few additional training steps on examples from these classes. This indicates that learning may be transferable between distinct types of tasks, allowing to explore transfer learning strategies.

This result is a step towards the development of optimization methods that frees the designer from tuning control parameters as it will be motivated in Section 2. The remaining parts of this paper are organized as follows: Section 3 presents the classical L-BFGS algorithm and discuss some methodologies to determine step sizes; Section 4 contains the architecture for the proposed policy and also discussions on how it was implemented; Section 5 describes the training procedure; and, finally, Section 6 presents experiments using classifiers to operate on MNIST and CIFAR-10 databases. The notation is mainly standard. Scalars are plain lower-case letters, vectors are bold lower-case, and matrices are bold upper-case. The clip function is defined as $\mathrm{clip}_l^u(y) := \min(u, \max(l, y))$.

## 2 MOTIVATION

Most algorithms used in artificial intelligence and statistics are based on optimization theory, which has widely collaborated for the success of machine learning applications in the last decades. However, this two-way bridge seems not to be currently leveraging its full potential in the other sense, that is, to learn how to automate optimization procedures. Indeed, performing satisfactory optimization, or solving learning problems, still relies upon the appropriate tuning of parameters of the chosen algorithm, which are often grouped with other hyper-parameters of the learning task. Despite the existence of several methodologies to obtain good values for these parameters (Bengio, 2000; Bergstra et al., 2011; Bergstra and Bengio, 2012; Snoek et al., 2015; Daniel et al., 2016; Dong et al., 2018), the search for tuning-free algorithms that perform better than heuristically designed ones is of great interest among practitioner and theoreticians. Indeed, besides the generally-desirable faster convergence, the ready-to-use nature of such algorithms allows the user to focus his attention on other problem-level hyper-parameters while the optimization procedure is automatically performed, resulting in better time and effort allocation. As recent advancements of machine learning have helped automatize the solution of numberless problems, optimization theory should equally benefit from these, balancing the bridge flows.

From a wider viewpoint, most optimization problem requires the user to select an algorithm and tune it to some extent. Although intuition and knowledge about the problem can speed-up these processes, trial-and-error methodologies are often employed which can be a time-consuming and inefficient task. With that in mind, the concept of *Learned optimizers* has been gathering attention in the last few years and, basically, refers to optimization policies and routines that were learned by looking at instances of optimization problems, here called *tasks*. This idea was introduced by Li and Malik (2016) and Andrychowicz et al. (2016) building upon previous results of "learning to learn" or "meta-learning" (Thrun and Pratt, 1998; Hochreiter et al., 2001). In the former, the authors presented an optimization policy based on a neural network trained by reinforcement learning and taking as input the history of gradient vectors at previous iterations. The latter adopts a long short-term memory (LSTM) to achieve a similar task, but the learning is done by truncated backpropagation through time after unrolling the proposed optimizer for a certain number of steps. Subsequently, it was shown in Metz et al. (2019) how multilayer perceptrons (MLP), adequately trained using a combined gradient estimation method, can perform faster in wall-clock time compared to current algorithms of choice. Also within this scenario, in Xu et al. (2019) a reinforcement learning-based methodology to auto-learn an adaptive learning rate is presented. Following this same fashion, in this present paper, instead of completely learning an optimizer from data, we propose a mixture of these ideas into a classical optimization procedure. Thus, the resulting optimizer, composed by a combination of L-BFGS and the proposed policy, will be learned in a constrained domain that assures valuable mathematical properties. The idea is to leverage both frameworks, inheriting the theoretical aspects assured by optimization theory while learning a policy to rule out the hand-design of parameters.

---

**Algorithm 1:** L-BFGS algorithm

---

**Input:** $s_i = x_{i+1} - x_i$, $y_i = g_{i+1} - g_i$ and $\rho_i = 1/(s_i^T y_i)$ for all $i \in k - m, \dots, k - 1$; and current gradient $g_k$,

**Result:** update direction $d_k = -H_k g_k$

---

1 $\quad q \leftarrow g_k$;
2 $\quad$ **for** $i = k - 1, \dots, k - m$ **do**
3 $\qquad \alpha_i \leftarrow \rho_i s_i^T q$;
4 $\qquad q \leftarrow q - \alpha_i y_i$;
5 $\quad$ **end**
6 $\quad \gamma = |s_{k-1}^T y_{k-1}| / (y_{k-1}^T y_{k-1})$ ;

7 $\quad r \leftarrow \gamma q$;
8 $\quad$ **for** $i = k - m, \dots, k - 1$ **do**
9 $\qquad \beta \leftarrow \rho_i y_i^T r$;
10 $\qquad r \leftarrow r + s_i(\alpha_i - \beta)$;
11 $\quad$ **end**
12 $\quad d_k \leftarrow -r$;

---

## 3 L-BFGS Algorithm

The L-BFGS algorithm was originally presented in Liu and Nocedal (1989) and is here transcribed into Algorithm 1. It is a quasi-Newton method derived from the BFGS algorithm (Nocedal and Wright, 2006) lowering space complexity from quadratic to linear in the problem dimension at the expense of precision. This algorithm calculates a descending direction in the search space taking into account an estimate of the inverse hessian matrix of $f(x)$, given by $H_k$. This matrix is not explicitly constructed but rather the product $d_k := -H_k g_k$ is obtained from the past $m$ values of $x_k$ and $g_k$, which have to be stored. This property makes it often the algorithm of choice for large-scale deterministic non-linear optimization problems. If $f(x)$ is convex in $x$, this algorithm is guaranteed to provide a descending update direction, but the same does not apply for non-convex objective functions. However, a simple way to circumvent this is by removing iterations $i$ in lines 2 and 8 of Algorithm 1 such that $\rho_i \leq 0$ (Nocedal and Wright, 2006, p. 537), which is used in this paper.

A matter of great relevance within this scope is how to choose an appropriate step size $t_k$ to apply the update rule in Eq. (2). To the best of our knowledge, there does not seem to exist a consensus on how to choose $t_k$ in a general way for non-convex objective functions. The scaling factor $\gamma$ in lines 6-7 of Algorithm 1 is known to assure that the step size $t_k = 1$ is accepted in most iterations in the convex optimization context, but not always. We will refer to a constant step-size policy that outputs $t_k = 1$ as the *baseline L-BFGS*. However, a *line search* (LS) procedure is often combined with L-BFGS to assure its convergence. Ideally, this should be performed by solving $t_k = \arg\min_{t>0} f(x_k + t d_k)$ but this exact approach is often too expensive to be adopted, motivating the use of inexact ones. An example is the *backtracking line search* (BTLS), which takes an initial length $t_k$ for the step size and shrinks it repeatedly until the so-called sufficient decrease Wolfe Condition $f(x_k + t_k d_k) \leq f(x_k) + c_1 t_k g_k^T d_k$ is fulfilled, where $c_1 \in (0, 1)$ is a control parameter to be tuned. Another parameter that has to be designed is the contraction factor $c_2 \in (0, 1)$ that shrinks the step size, i.e., $t_k \leftarrow c_2 t_k$, see Nocedal and Wright (2006, p. 37). This method assures convergence to a local-minima at the cost of re-evaluating the objective function several times per iteration. This is a price that the user is, in some cases, willing to pay, but for large-dimensional problems this procedure is likely to become the bottle-neck of the optimization task. It is important to highlight that the method to be presented may also apply to other optimization algorithms that deeply rely on line searches to perform well. However, this paper focus on L-BFGS as it is often the algorithm of choice in large-scale deterministic optimization.

In the context of stochastic optimization, many modified versions of Algorithm 1 together with methodologies for choosing $t_k$ are available (Moritz et al., 2016; Zhou et al., 2017; Bollapragada et al., 2018; Wills and Schön, 2019), but for sake of simplicity, our work will deal exclusively with deterministic non-linear optimization problems.

## 4 Learned policy for selecting step sizes

Recalling the definition of $s_k$ and $y_k$ in Algorithm 1, our policy is defined as $t_k = \pi(d_k, g_k, s_{k-1}, y_{k-1}; \theta)$ and selects an adequate step size for L-BFGS but neither relying on any parameter tuning nor requiring additional evaluations of the objective function. Instead, its parameters that are represented by $\theta$ should be learned from data. Let us, from now on, de-

$\pi(\cdot;\boldsymbol{\theta})$

$\boldsymbol{d}_k, \boldsymbol{g}_k, \boldsymbol{s}_{k-1}, \boldsymbol{y}_{k-1}$ → $\mathbf{dotln}(\cdot)$ → $\boldsymbol{u}_0$

Linear Layer 1 ($\mathbb{R}^{16} \to \mathbb{R}^{n_h}$) → $\boldsymbol{u}_1$

Linear Layer 2 ($\mathbb{R}^{16} \to \mathbb{R}^{n_h}$) → $\boldsymbol{u}_2$

$p(\cdot)$ ($\mathbb{R}^{n_h} \to \mathbb{R}$) → $\tau_k$ → $\exp(\cdot)$ → $t_k$

Figure 1: Neural network architecture for the proposed policy to generate step sizes $t_k$.

fine this policy combined with Algorithm 1 as the *L-BFGS-$\pi$ approach*. The architecture of the policy $\pi(\boldsymbol{d}_k, \boldsymbol{g}_k, \boldsymbol{s}_{k-1}, \boldsymbol{y}_{k-1}; \boldsymbol{\theta})$ is shown in Fig. 1. To allow the policy to be independent from the problem size $n$, only the inner products between its inputs are used. These values define $\boldsymbol{u}_0 = \mathbf{dotln}(\boldsymbol{d}_k, \boldsymbol{g}_k, \boldsymbol{s}_{k-1}, \boldsymbol{y}_{k-1})$ where $\mathbf{dotln}(\cdot)$ returns the component-wise application of $f(x) = \ln(\min(x, \epsilon))$ to the elements of $\boldsymbol{X} = [\boldsymbol{d}_k\ \boldsymbol{g}_k\ \boldsymbol{s}_{k-1}\ \boldsymbol{y}_{k-1}]^T[\boldsymbol{d}_k\ \boldsymbol{g}_k\ \boldsymbol{s}_{k-1}\ \boldsymbol{y}_{k-1}]$ but with the superdiagonal entries having their signs reversed. We have chosen $\epsilon = 10^{-8}$ to avoid imaginary-valued entries.

The vector $\boldsymbol{u}_0$ is the input to two parallel input layers, which are fully connected linear layers that transport information in $\boldsymbol{u}_0$ to another vector space $\mathbb{R}^{n_h}$ (in our tests, we adopted $n_h = 6$). Their outputs, as usual, are defined as $\boldsymbol{u}_1 = \boldsymbol{W}_{01}\boldsymbol{u}_0 + \boldsymbol{b}_{01}$ and $\boldsymbol{u}_2 = \boldsymbol{W}_{02}\boldsymbol{u}_0 + \boldsymbol{b}_{02}$. The logarithm operation was adopted to let the linear layers evaluate products and divisions between powers of the inputs by simply summing and subtracting them. Moreover, as the output is positive, working in the logarithmic vector space allows us to use a wider range of numerical values. Subsequently, let us define the normalized vectors $\bar{\boldsymbol{u}}_1 = \boldsymbol{u}_1/\|\boldsymbol{u}_2\|$ and $\bar{\boldsymbol{u}}_2 = \boldsymbol{u}_2/\|\boldsymbol{u}_2\|$ to calculate the scalar projection of $\bar{\boldsymbol{u}}_1$ onto $\bar{\boldsymbol{u}}_2$ and clip the result to some interval $[\tau_m, \tau_M]$, yielding the *log-step size*

$$\tau_k = \mathrm{clip}_{\tau_m}^{\tau_M}\left(\bar{\boldsymbol{u}}_2^T \bar{\boldsymbol{u}}_1\right) =: p(\boldsymbol{u}_1, \boldsymbol{u}_2) \tag{3}$$

Finally, the selected step size is obtained as $t_k = e^{\tau_k}$. To geometrically interpret this, we sketch three different scenarios in Fig. 2. The dashed lines represent orthogonal axes spanned by some arbitrary $\bar{\boldsymbol{u}}_2$ and the gray strip represents the interval $[\tau_m, \tau_M]$ along the direction of $\bar{\boldsymbol{u}}_2$ whence $\tau_k$ should be taken. When the Linear Layer 1 maps $\boldsymbol{u}_0$ into $\boldsymbol{u}_1'$, the scalar projection of $\bar{\boldsymbol{u}}_1'$ onto $\bar{\boldsymbol{u}}_2$ is beyond the maximal $\tau_M$, so $\tau_k$ is clipped to it. In the same way, for $\bar{\boldsymbol{u}}_1'''$ the step size will be the minimal one $t_k = e^{\tau_m}$ whereas for the intermediate $\bar{\boldsymbol{u}}_1''$ we have $\tau_k \in (\tau_m, \tau_M)$. The two layers, jointly trained, should learn how to position $\bar{\boldsymbol{u}}_1$ and $\bar{\boldsymbol{u}}_2$ in the lifted space to represent important directional information of $\boldsymbol{d}_k$ and $\boldsymbol{g}_k$ by looking at similar optimization tasks, being thus able to produced suitable step sizes.

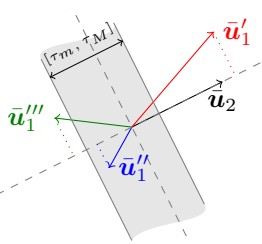

Figure 2: Geometric representation of the scalar projection and clip procedures for 3 cases.

This approach is powerful enough to capture interesting mathematical local properties of this problem. As an instance, it could calculate $\cos\phi_k = -\boldsymbol{d}_k^T\boldsymbol{g}_k/\sqrt{\boldsymbol{d}_k^T\boldsymbol{d}_k\boldsymbol{g}_k^T\boldsymbol{g}_k}$, where $\phi_k$ is the angle formed between $\boldsymbol{d}_k$ and the steepest descend direction $-\boldsymbol{g}_k$, by letting parameters $\boldsymbol{\theta} := (\boldsymbol{W}_{01}, \boldsymbol{b}_{01}, \boldsymbol{W}_{02}, \boldsymbol{b}_{02})$ and limits $\tau_m$ and $\tau_M$ be given as described in Appendix A. Also, $\boldsymbol{s}_{k-1}^T\boldsymbol{y}_{k-1}$ defines the so-called *curvature condition* (Nocedal and Wright, 2006, p. 137) and the occurrence of small-valued $\boldsymbol{g}_k^T\boldsymbol{g}_k$ and $\boldsymbol{s}_{k-1}^T\boldsymbol{s}_{k-1}$ with $\boldsymbol{s}_{k-1}^T\boldsymbol{y}_{k-1} < 0$ implies that the iterate $\boldsymbol{x}_k$ approximates a local maximum or a saddle point, as this inequality indicates that at least one eigenvalue of the hessian matrix is negative at $\boldsymbol{x}_k$.

Indeed, the considered inner products forming $\boldsymbol{u}_0$ are also employed in many procedures for determining step sizes, for example, in the sufficient decrease Wolfe condition for backtracking line search, which makes our policy comparable to them in the sense that $\pi(\cdot; \boldsymbol{\theta})$ does not require additional information to operate.

However, notice that the clip function is not suitable for training given that it is non-differentiable and gradients cannot be backpropagated through it. Fortunately, the clip operation (3) can be cast as

a convex optimization problem

$$\tau_k = \arg\min_{\tau \in \mathbb{R}} \|\boldsymbol{u}_2 \tau - \boldsymbol{u}_1\|^2 \tag{4}$$

$$\text{s.t. } \tau_m \leq \tau \leq \tau_M \tag{5}$$

allowing $\tau_k$ to be calculated by a convex optimization layer, defined here as a *CVX Layer*, (Agrawal et al., 2019). This last layer can output the solution to a parameter-dependent convex optimization problem. For the special case where a solution is not differentiable with respect to the input (e.g., in our case when an inequality constraint is active), the automatic differentiation procedure delivers an heuristic quantity that can be employed as a gradient. The use of a CVX Layer is therefore convenient for training our policy but, on the other hand, using Eq. (3) in its place when applying the already-trained policy significantly speeds up the step-size evaluation, compared to solving (4).

It is important to remark that this policy is defined as independent from both the memory length $m$ of Algorithm 1 and the problem dimension $n$. Additionally, the lower and upper limits for the log-step size are $\tau_m$ and $\tau_M$, respectively, and can also be learned. In this work, however, we chose $\tau_m = -3$ and $\tau_M = 0$, letting $t_k \in [0.0497, 1]$. This interval is comprehensive enough to let our method be compared in a fair way to backtracking line searches. Moreover, when we allowed $\tau_M$ to be learned in our tests it converged to values that were very close to $\tau_M = 0$, indicating that 1 was already an adequate upper limit for the step size.

## 5 Training the policy

The L-BFGS-$\pi$ procedure can be trained by *truncated backpropagation through time* (TBPTT), in a similar way to Andrychowicz et al. (2016). From this point on, training the optimizer is referred to as the *outer optimization problem* whereas an instance of a task in the form of (1) is called the *inner optimization problem*. Therefore, this outer problem is defined as

$$\underset{\boldsymbol{\theta}}{\text{minimize}} \, F(\boldsymbol{\theta}) := \mathbb{E}_{\boldsymbol{x}_0 \sim \mathbb{R}^n} \mathbb{E}_{f \sim \mathcal{T}} \left( \sum_{k=1}^{K} w_k f(\boldsymbol{x}_k) \right) \tag{6}$$

$$\text{s.t. } \boldsymbol{x}_{k+1} = \boldsymbol{x}_k + \pi(\boldsymbol{d}_k, \boldsymbol{g}_k, \boldsymbol{s}_{k-1}, \boldsymbol{y}_{k-1}; \boldsymbol{\theta}) \boldsymbol{d}_k \tag{7}$$

where $\boldsymbol{d}_k$ is given by Algorithm 1, $K \in \mathbb{N}$ is the truncated horizon over which optimization steps are unrolled, $w_k$, $k = 1, \ldots, K$ are weight scalars, herein considered $w_k = 1$, and $\mathcal{T}$ is some set of tasks formed by inner objective functions $f(\boldsymbol{x})$ to be optimized. In (6), the innermost expected value is approximated by sampling tasks within a training set $\mathcal{T}_{train}$, one at a time, and unrolling the optimization for $K$ inner steps for some random $\boldsymbol{x}_0$ with i.i.d. components. One outer optimization step consists of, performing $K$ inner steps, computing a gradient for the outer optimization problem $\nabla_{\boldsymbol{\theta}} F(\boldsymbol{\theta})$ and updating $\boldsymbol{\theta}$, in our case, by ADADELTA with learning rate equals 1 (Zeiler, 2012). To assure that different orders of magnitude of $\boldsymbol{x}$ are seen during this training, we set the initial point for the next outer step to be the last iterate from the previous one, i.e., $\boldsymbol{x}_0 \leftarrow \boldsymbol{x}_K$, and perform another whole outer optimization step. This is repeated for $T$ outer steps or until $\|\boldsymbol{g}_k\| < \epsilon = 10^{-10}$, when a new random $\boldsymbol{x}_0$ is then sampled. Backpropagation to calculate $\nabla_{\boldsymbol{\theta}} F(\boldsymbol{\theta})$ happens through all operations with exception of the inner gradient evaluation $\boldsymbol{g}_k$, which is considered an external input. Double floating-point precision is used to assure accurate results in the comparisons.

## 6 Example: Training a classifier on MNIST

All tests were carried out with the aid of `PyTorch` (Paszke et al., 2019) to backpropagate gradients and of `cvxpylayers` (Agrawal et al., 2019) to implement the CVX layers. They were run on an Intel Xeon Gold 6132 equipped with an NVidia V100. First, we carried out tests on convex optimization problems, namely quadratic functions and logistic regression problems, but no significant difference was noticed and these were, therefore, omitted.

In this example, we trained an MLP with $n_l = 1$ hidden layer with $n_u = 20$ units and sigmoid activation functions to classify images of digits in MNIST database (LeCun et al., 1998). This model is adapted from one example in Andrychowicz et al. (2016). We used a full-batch gradient at every iteration, even though stochastic optimization is generally the most common strategy employed in similar tasks. Nevertheless, our main interest in this example is not the classification problem itself but rather to analyze the optimization problem and how our deterministic algorithm performs on it.

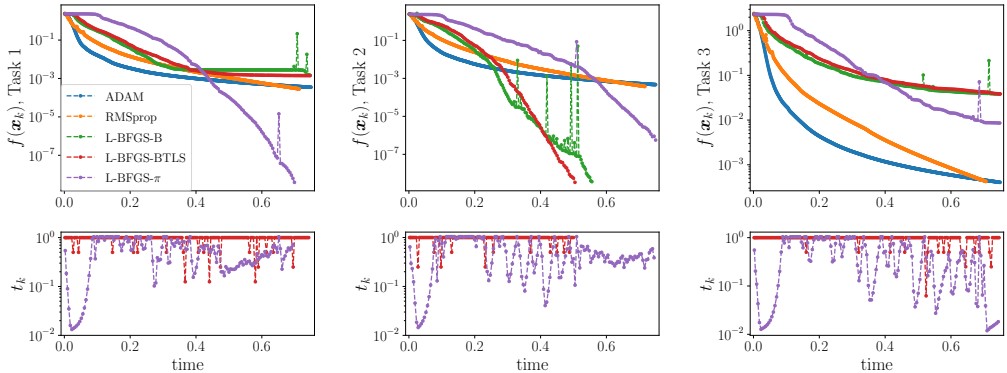

Figure 3: Objective function $f(\boldsymbol{x}_k)$ at the current iterate with respect to wall-clock time for all optimizers and 3 selected tasks (top) and correspondent step sizes for $\pi(\cdot, \boldsymbol{\theta})$ and BTLS (bottom).

The $n = 16{,}280$ parameters defining an MLP are concatenated in $\boldsymbol{x}$ and $f(\boldsymbol{x})$ is the associated cross-entropy loss function for a given set of images. This is known to be a non-convex optimization problem, mainly because of the presence of non-linear activation functions. A training set of tasks $\mathcal{T}_{train}$ was constructed by randomly grouping images in MNIST training set into 60 batches of $N = 1{,}000$ images. For each of these batches one initial condition $\boldsymbol{x}_0$ was sampled, which altogether compose $\mathcal{T}_{train}$ with 60 tasks. The policy $\pi(\cdot; \theta)$ was trained for 50 epochs, $K = 50$ and $T = 8$, and its performance was compared to other methods, namely, ADAM, RMSProp, L-BFGS with a BTLS and the baseline L-BFGS (referred to as L-BFGS-B). For running Algorithm 1, we selected $m = 5$. The learning rates of ADAM and RMSProp were heuristically tuned to yield fast convergence by exhaustive search within the set $\left\{i \times 10^j \ : \ i \in \{1,3\}, \ j \in \{-3, \ldots, -1\}\right\}$ and the values 0.03 and 0.01 were used, respectively. The BTLS parameters $c_1$ and $c_2$ were searched in the set $\{0.25, 0.5, 0.75\}^2$ and $c_1 = 0.25$, $c_2 = 0.5$ were chosen, associated to the best results, i.e., fastest convergence on tasks in $\mathcal{T}_{train}$. The initial step size for the BTLS was $t_k = 1$. The following comparisons were performed in a test set of tasks $\mathcal{T}_{test}$ built similarly to $\mathcal{T}_{train}$ but considering all images in the MNIST test set split into 10 batches of $N = 1{,}000$ images, and 100 random samples of $\boldsymbol{x}_0$ were generated for each batch, resulting in 1,000 tasks. The optimization was performed for $K = 800$ steps or until $\|\boldsymbol{g}_k\| < 10^{-8}$. The first 3 samples for each optimizer were considered "warm-up" runs and, therefore, were discarded to avoid having time measurement affected by any initial overhead.

The objective function value for three selected tasks is shown in the upper plots of Fig. 3 along with the correspondent selected step sizes by $\pi(\cdot; \boldsymbol{\theta})$ and the BTLS, on the bottom ones. For Task 1, L-BFGS-$\pi$ was successful in attaining lower values for $f(\boldsymbol{x})$ when compared to the other algorithms. For some tasks, such as Task 2, poorer performance was noticed when compared to the other L-BFGS approaches and, for some other tasks as Task 3, Adam and RMSprop are more successful than the others. This suggests that none of these methods outperforms the others in a general case. Notice that in these figures, the spikes in the curves associated with L-BFGS-$\pi$ and with the baseline L-BFGS-B represent steps at which $t_k = \pi(\cdot; \boldsymbol{\theta})$ and $t_k = 1$, respectively, were not step sizes that provided a decrease. Results for other tasks are presented in Appendix B.

For each individual task, the first instant of time $t_f$ at which the optimization procedure attained some precision-based stop criteria $\|\boldsymbol{g}_k\| < \varepsilon$ for different values of $\varepsilon$ was computed for all four optimization procedures, and a comparison between our methodology and others is shown in Fig. 4. These plots compare algorithms two-by-two and the way to interpret them is by observing that a point above the blue line represents a task for which the algorithm associated to the $x$ axes reached the precision criterion faster, and *vice-versa*. If such $t_f$ does not exist, we define $t_f = \infty$ and the two subplots, on the right and on the top, are use to represent tasks for which the given precision was reached by one of the algorithms but not by the other. Tasks for which the criterion was not reached by both algorithms are not displayed. Notice that better precision values were reached by our approach when compared to ADAM and RMSProp but a similar performance was obtained

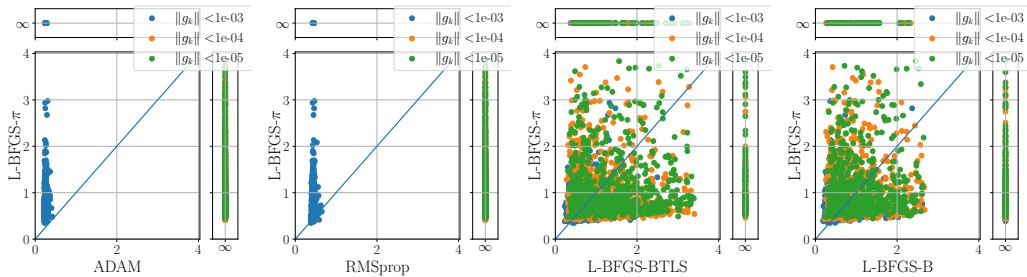

Figure 4: Time $t_f$ (in seconds) at when the step criterion $\|g_k\| < \varepsilon$ was reached for all tasks in $\mathcal{T}_{test}$ compared for different optimizers and $(n_l, n_u) = (1, 20)$.

Table 1: Percentages of wins (W) and ties (T) of L-BFGS-$\pi$ trained for $(n_l, n_u) = (1, 20)$ against other algorithms on tasks in $\mathcal{T}_{test}$ with respect to which one attained the stop criterion $\|g_k\| < \varepsilon$ first for different $(n_l, n_u)$.

| Competitor | $(n_l, n_u) = (1, 20)$ | | | $(n_l, n_u) = (2, 400)$ | | | $(n_l, n_u) = (4, 800)$ | | |
|---|---|---|---|---|---|---|---|---|---|
| | $\varepsilon$ | W (%) | T (%) | $\varepsilon$ | W (%) | T (%) | $\varepsilon$ | W (%) | T (%) |
| ADAM | $10^{-3}$ | 0 | 0 | $10^{-5}$ | 4.7 | 0 | $10^{-5}$ | 0 | 0 |
| | $10^{-4}$ | 95.8 | 4.2 | $10^{-6}$ | 5.8 | 0 | $10^{-6}$ | 0.1 | 0 |
| | $10^{-5}$ | 90.9 | 9.1 | $10^{-7}$ | 10.5 | 0 | $10^{-7}$ | 0.2 | 0.1 |
| RMSProp | $10^{-3}$ | 5.5 | 0 | $10^{-5}$ | 100 | 0 | $10^{-5}$ | 92.7 | 7.3 |
| | $10^{-4}$ | 95.8 | 4.2 | $10^{-6}$ | 100 | 0 | $10^{-6}$ | 81.0 | 19.0 |
| | $10^{-5}$ | 90.9 | 9.1 | $10^{-7}$ | 100 | 0 | $10^{-7}$ | 67.2 | 32.8 |
| L-BFGS-B | $10^{-3}$ | 32.1 | 0 | $10^{-5}$ | 5.7 | 0 | $10^{-5}$ | 9.2 | 1.2 |
| | $10^{-4}$ | 49.3 | 0.5 | $10^{-6}$ | 5.5 | 0 | $10^{-6}$ | 8.5 | 2.1 |
| | $10^{-5}$ | 54.9 | 3.3 | $10^{-7}$ | 5.5 | 0 | $10^{-7}$ | 7.6 | 3.0 |
| L-BFGS-BTLS | $10^{-3}$ | 40.8 | 0 | $10^{-5}$ | 1.0 | 0 | $10^{-5}$ | 0 | 0 |
| | $10^{-4}$ | 55.6 | 0.6 | $10^{-6}$ | 1.0 | 0 | $10^{-6}$ | 0 | 0 |
| | $10^{-5}$ | 59.0 | 2.5 | $10^{-7}$ | 1.2 | 0 | $10^{-7}$ | 0 | 0 |

when compared to the heuristically designed backtracking line search method, which is the gold standard.

Additionally, Table 1 presents the percentage of times that L-BFGS-$\pi$ reached the defined precision before other methods, characterizing a "win", and that both methods reached the precision at the exact same time (which is very unlikely) or have not reached this precision after $K = 800$ inner steps, denoting a "tie". To investigate whether our policy is able to generalize and perform well on higher-dimension problems, we also present these results for $(n_l, n_u)$ equals to $(2, 400)$ and $(4, 800)$, characterizing problems of size $n = 637,600$ and $n = 128,317,600$ respectively. Different values of $\varepsilon$ were considered as smaller values for $\|g_k\|$ were reached for these two last cases.

For $(n_l, n_u) = (1, 20)$, which contains problems of the same dimension as those seen during training, L-BFGS-$\pi$ clearly outperforms RMSProp and ADAM whereas it presents a slightly faster convergence than L-BFGS-B and L-BFGS-BTLS for smaller $\varepsilon$. Unfortunately, for higher dimension problems, the proposed policy did not achieve the same level of performance as in the problem it was trained for.

In spite of that, given the non-convexity of this problem, it is also important to observe what were the minimum values obtained for $f(x)$ by each algorithm. As the proposed policy does not assure a decreasing step size at each iteration, instead of the final value $f(x_K)$ we looked at $f_* := \min_k f(x_k)$, which can easily be stored and updated during the optimization. However an analogous discussion is presented in Appendix C but considering only the final values $f(x_K)$ and the same conclusions are drew. More than simply looking at the minimum values, we would like to verify whether L-BFGS-$\pi$

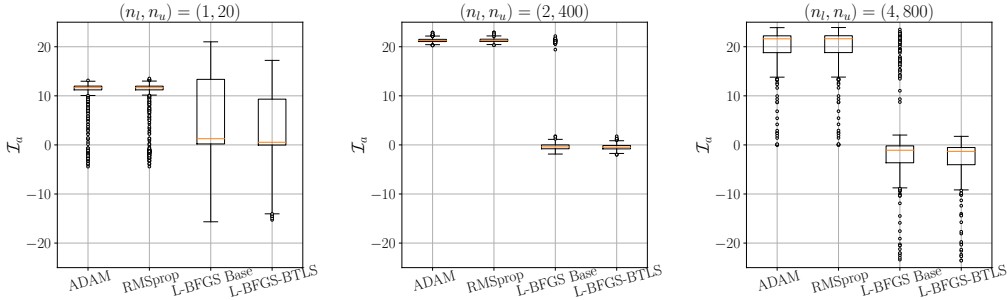

Figure 5: Box plots for values of $\mathcal{I}_a$, where $a$ is the algorithm in the $x$-axis, encountered for all tasks in $\mathcal{T}_{test}$ by different algorithms and pairs $(n_l, n_u)$.

attains lower values $f_*$ when compared to other algorithms. To this end we present the index

$$\mathcal{I}_a(f) = \ln\left(f_*^{[a]}/f_*^{[\text{L-BFGS-}\pi]}\right) \tag{8}$$

where $f_*^{[a]}$ represents the minimum value reached by some algorithm $a$ for $f(\boldsymbol{x})$. Hence, $\mathcal{I}_a(f) > 0$ implies that L-BFGS-$\pi$ performs better than $a$ in the task associated to $f(\boldsymbol{x})$ and its initial condition. Box plots of the obtained values for all tasks and each one of the other algorithms are presented in Fig. 5. In these plots we can notice that L-BFGS-$\pi$ reaches, in average, better values than all of its competitors. Also, ADAM and RMSprop generalized very poorly to higher-dimension problems, indicating that some re-tuning is required for these algorithms. Under this perspective, L-BFGS-$\pi$ had a similar performance to L-BFGS-BTLS and L-BFGS-B, despite the presence of some outliers indicating cases where our policy reached bad local minima. This showed how the proposed policy was successful in learning to provide step sizes in a single shot that are as good as those generated by a heuristically designed line search, which benefits from the possibility of re-evaluating the objective function as much as needed.

Finally, as a last experiment, we applied the learned policy and these competitors to a class of tasks comprising the training of a Convolutional Neural Network (CNN) to classify images in CIFAR-10, see (Krizhevsky et al., 2009). The adopted architecture is described in Zhang (2016) but sigmoid activation functions were replaced by ReLU to make this problem even more distant from the one $\pi$ was trained on. A training and a test set of tasks, $\mathcal{T}_{train}^C$ and $\mathcal{T}_{test}^C$, were built similarly to $\mathcal{T}_{train}$ and $\mathcal{T}_{test}$ but using images in CIFAR-10 instead. Evaluating these algorithms in $\mathcal{T}_{test}^C$ and computing the index $\mathcal{I}_a(f)$ for each task allows us to present the first box plot in Fig. 6. This figure indicates that $\pi$ do not perform as good as before in these problems. This could be expected as a different architecture directly affects the nature of the objective function. To investigate whether the learned policy $\pi$ can be used as a warm-start for the training a new policy $\pi^C$, we perform additional training steps on $\pi$ corresponding to 10 epochs in the training set $\mathcal{T}_{train}^C$, but eliminating 5/6 of its tasks. This is done to show that even with very low effort placed in this retraining phase and considering fewer learning data, we can benefit from previous learning to speed-up new training procedures. The corresponding results are presented in the second box plot of Fig. 6, which shows that the new policy $\pi^C$ performs comparably to the competitors. Certainly, further investigation is required but this suggests that some learning can be transferred across distinct problem domains.

## 7 CONCLUSIONS

In this work we demonstrate how to build and train a neural network to work as step-size policy for the L-BFGS algorithm. The step sizes provided by this policy are of the same quality as those of a backtracking line searches, hence making the latter superfluous. Moreover, L-BFGS with our step-size policy outperforms, in wall-clock time and optimal/final value, ADAM and RMSprop with heuristically tuned parameters in training classifiers for the MNIST database. Also, we showed how a learned policy can be used to warm start the training of new policies to operate on different classes of problems. In future work, we intend to extend this result for stochastic optimization,

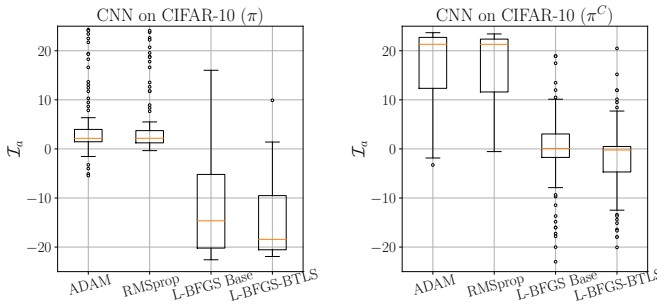

Figure 6: Box plots for values of $\mathcal{I}_a$, where $a$ is the algorithm in the $x$-axis, encountered for all tasks in $\mathcal{T}_{test}^C$ by different algorithms when compared against $\pi$ and $\pi^C$ on the training of a CNN.

allowing us to learn policies to determine, for example, learning rates in other classic machine learning algorithms.

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

## A   PARAMETERS FOR THE POLICY TO CALCULATE $\cos \phi_k$

In Section 4, it was stated that the policy $\pi(\cdot; \boldsymbol{\theta})$ is able to calculate

$$\cos \phi_k = \frac{-\boldsymbol{d}_k^T \boldsymbol{g}_k}{\sqrt{\boldsymbol{d}_k^T \boldsymbol{d}_k \boldsymbol{g}_k^T \boldsymbol{g}_k}}. \tag{9}$$

In the simplest case, this can be done by choosing $\boldsymbol{W}_{01}$, $\boldsymbol{b}_{01}$, $\boldsymbol{W}_{02}$ and $\boldsymbol{b}_{02}$ adequate to make $\boldsymbol{u}_1 = \ln(-\boldsymbol{d}_k^T \boldsymbol{g}_k) - 0.5 \ln(\boldsymbol{d}_k^T \boldsymbol{d}_k) - 0.5 \ln(\boldsymbol{g}_k^T \boldsymbol{g}_k)$ and $\boldsymbol{u}_2 = 1$. The optimization problem becomes

$$\tau_k = \arg \min_{\tau \in \mathbb{R}} \left( \tau - \ln \left( \cos \phi_k \right) \right)^2 \tag{10}$$

$$\text{s.t. } \tau_m \leq \tau \leq \tau_M \tag{11}$$

Recalling that $\cos \phi_k > 0$ as $-\boldsymbol{d}_k^T \boldsymbol{g}_k > 0$, letting $\tau_M = 0$ and $\tau_m$ small enough assures that $t_k = e^{t_k} = \cos \phi_k$. It is important to say that this specific step size might not be a good one, but this quantity can carry useful information when composing vectors $\boldsymbol{u}_1$ and $\boldsymbol{u}_2$ as it characterizes the deviation between the update direction $\boldsymbol{d}_k$ and the steepest descend direction.

## B   SAMPLES OF TEST TASKS

In this appendix we provide in Fig. 7 the objective function $f(\boldsymbol{x}_k)$ obtained in our tests for the 10 first tasks in $\mathcal{T}_{test}$. Differently form the results in Fig. 3 that were chosen by inspection, the plots in Figure 7 should represent a more uniform visualization of the policy behavior in this set.

## C   FINAL VALUE ANALYSIS

Here we present the results regarding the index $\mathcal{I}_a(f)$ defined in (8) but in the case where one chooses to define $f_*^{[a]}$ based on the final value $f(\boldsymbol{x}_K)$ obtained after applying the algorithm $a$ for $K = 800$ iterations. The box plot in Fig. 5 is reconstructed and presented in Fig. 8. The conclusion drawn from this analysis is the same as the one obtained in the former definition of $f_*^{[a]}$, based on the minimum value $f(\boldsymbol{x}_k)$ for all $k$. However, in the context of deterministic nonlinear optimization it is a good idea to keep the best visited iterate so far and allow the algorithm to explore other areas of the decision space. This is the reasoning that motivates considering the minimum over the iterations in the main analysis.

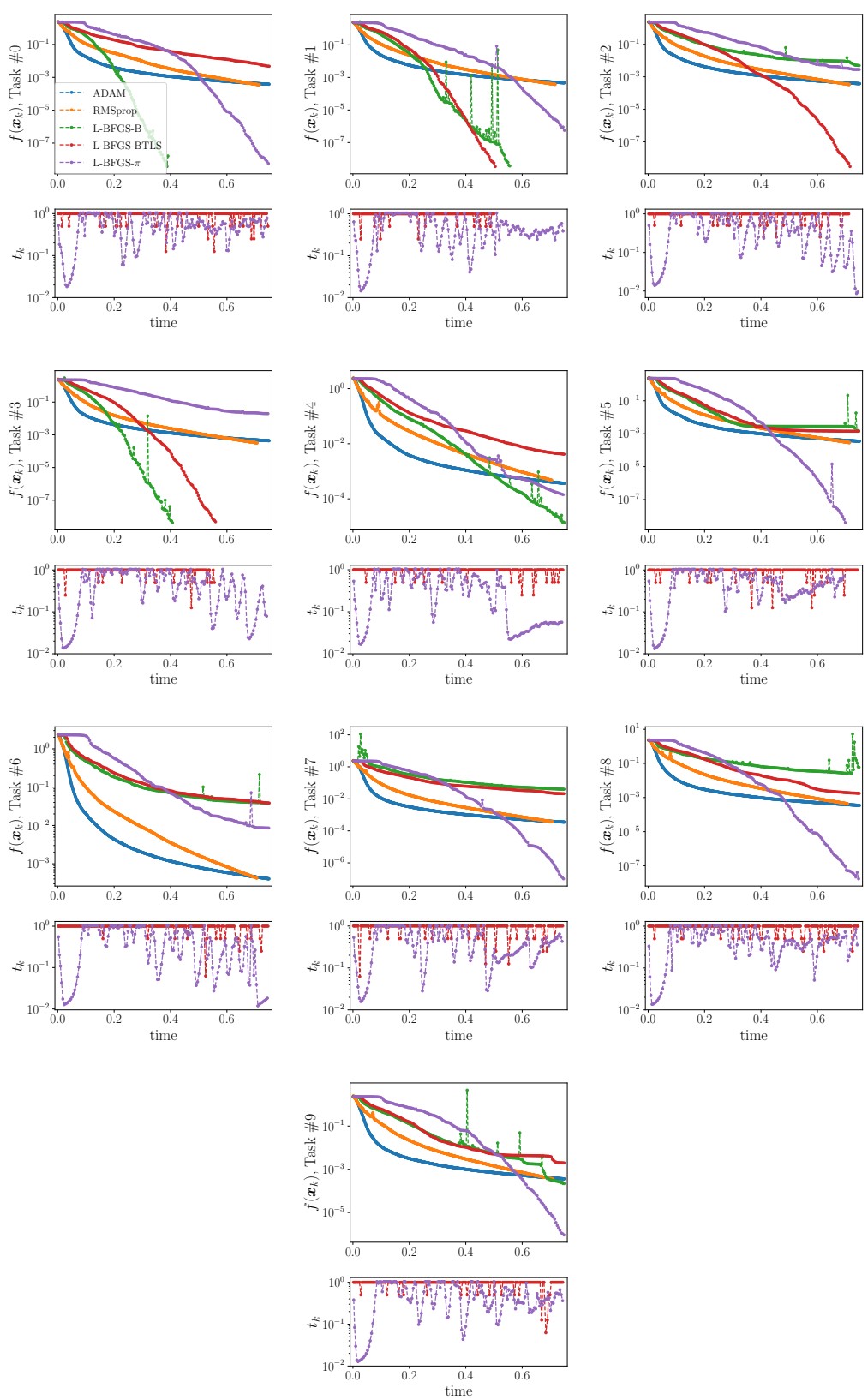

Figure 7: Objective function $f(\boldsymbol{x}_k)$ at the current iterate with respect to wall-clock time for all optimizers and the 10 first tasks in $\mathcal{T}_{test}$ (top) and correspondent step sizes for $\pi(\cdot, \boldsymbol{\theta})$ and BTLS (bottom).

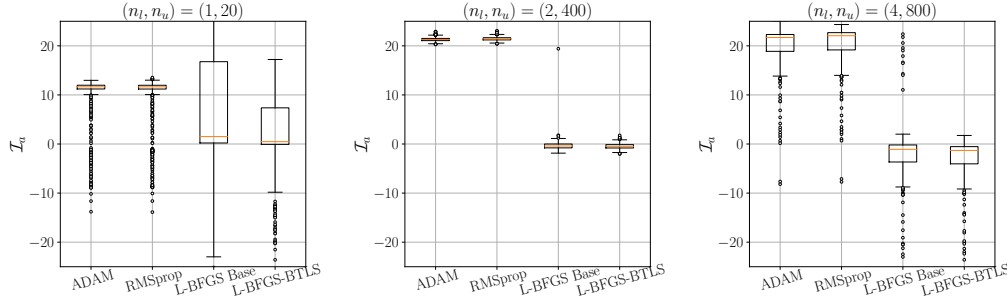

Figure 8: Box plots for values of $\mathcal{I}_a$, where $a$ is the algorithm in the $x$-axis, encountered for all tasks in $\mathcal{T}_{test}$ by different algorithms and pairs $(n_l, n_u)$ (for $f_*^{[a]}$ defined as the final value $f(\boldsymbol{x}_K)$ obtained by algorithm $a$).

