# OpenReview forum: "Learning the Step-size Policy for the Limited-Memory Broyden-Fletcher-Goldfarb-Shanno Algorithm"
_ICLR.cc/2021/Conference — Reject_

### Official Review · AnonReviewer1 · 2020-10-15
**Interesting submission about step-length policy learning**

**Rating:** 4
**Confidence:** 3

**Review:**

1. Paper contribution summary
    This paper proposes a neural network architecture with local gradient/quasi gradient as input to learn the step-length policy from data for the specific L-BFGS algorithm. The step-length policy can be learned from the data with similar optimization problems. The designed neural network architecture for step-length policy is using inner product as input to allow the policy to be independent of the problem size, and the logarithm operation is used to enable products and division among powers of the inputs.
    The numerical example shows that the learned policy is comparable to L-BFGS with backtracking line search and potentially better generalization to the same problem with higher dimension.

2.  Strong and weak points of the paper
    Strong part: 1) The network architecture for the step-length policy is very interesting, and may be useful for other problems with similar needs. 2) The numerical experiment shows that the learned step-length policy is comparable to the same algorithm with backtracking line searched step-length, which looks promising.
    Week part: 1) The paper's goal is limited to design the step-length policy for one specific optimization algorithm L-BFGS. If the original L-BFGS algorithm is not effective for certain problems, then the learned step-length policy may be un-useful as well. 2) Based on Table 1's metric of gradient length smaller than \epsilon, the learned step-length policy seems not better than Adam even in the higher dimension setup in which the Adam is using the pre-tuned learning rate. The result in Fig. 5 seems better for the learned step-length policy, but that uses the smallest value over training as compared value, which may not be 'fair'. Because Adam, RMSProp, and L-BFGS are different algorithms, which may result in different (local) minima in different problems. 3) The performance test setup is not realistic in neural network model training. The setup for training the model (inner optimization) along with step-length policy model (outer optimization) is with a fixed 1000 images (randomly selected from 60 batches of data) over T = 16 outer iteration and 10 epochs. When comparing different algorithms' performance, it also splits the test dataset into 10 batches of 1000 images. And record performance for each fixed batch dataset, which is not realistic in usual neural network model training. Usually, different batches are randomly selected at each time step as opposed to being fixed. This makes the claimed performance doubtful in realistic setting.

3. Based on the strong and week points of this paper, I tend to reject it.

4. Supporting arguments
    Please refer to my above arguments.

5. Questions
    1) Can author/s please explain the first two plots in Figure 4? It seems to me that both Adam and RMSProp are faster reaching to the required gradient norm precision. The precision seems different between these two algorithms and the author/s claimed one. Why not use the same precision requirement so that we can compare these algorithms directly?
    2) Can author/s provide any feedback on the third point of "weak part"?
    3) Can this proposed architecture/idea be used as a step-size policy for not just L-BFGS?

---

> ### Author Response · Authors · 2020-11-23
> **Author's response to AnonReviewer1**
>
> **Authors**
>
> We would like to thank the reviewer for evaluating our work and sharing
> her/his thoughts here.
>
> **Reviewer**
>
> _Week part: 1) The paper’s goal is limited to design the step-length
> policy for one specific optimization algorithm L-BFGS. If the original
> L-BFGS algorithm is not effective for certain problems, then the learned
> step-length policy may be un-useful as well._
>
> **Authors**
>
> The reviewer is correct that the paper focus in L-BFGS specifically but
> we would like to highlight that L-BFGS is generally the algorithm of
> choice in large-scale deterministic optimization problems. However,
> there are caveats concerning the step-size selection and this is the
> problem we tackled in this paper.
>
> **Reviewer**
>
> _2\) Based on Table 1’s metric of gradient length smaller than
> $\backslash$epsilon, the learned step-length policy seems not better
> than Adam even in the higher dimension setup in which the Adam is using
> the pre-tuned learning rate. The result in Fig. 5 seems better for the
> learned step-length policy, but that uses the smallest value over
> training as compared value, which may not be ’fair’. Because Adam,
> RMSProp, and L-BFGS are different algorithms, which may result in
> different (local) minima in different problems._
>
> **Authors**
>
> Notice that in the revised version of the paper the results are
> different from the previous. For the problem scale in which the policy
> was trained it performs better than the competitors but the same is not
> true for an arbitrary different problem. We tried to motivate that by
> carrying out an additional experiment. Also, regarding the analysis of
> the attained final value, a new discussion was added in Appendix C.
> However, notice that in the context of deterministic nonlinear
> optimization it is a good idea to keep the best iterate visited so far
> and allow the algorithm to explore other areas of the decision space.
> That is why the minimum over the iterations was taken into account. To
> be fair, the same criterion was used for the competitors.
>
> **Reviewer**
>
> _3\) The performance test setup is not realistic in neural network model
> training. The setup for training the model (inner optimization) along
> with step-length policy model (outer optimization) is with a fixed 1000
> images (randomly selected from 60 batches of data) over T = 16 outer
> iteration and 10 epochs. When comparing different algorithms’
> performance, it also splits the test dataset into 10 batches of 1000
> images. And record performance for each fixed batch dataset, which is
> not realistic in usual neural network model training. Usually, different
> batches are randomly selected at each time step as opposed to being
> fixed. This makes the claimed performance doubtful in realistic setting._
>
> **Authors**
>
> We agree with the reviewer that training using mini-batches is generally
> preferable in this context but notice that the goal of this evaluation
> is not to tackle state-of-the-art machine learning problems but rather
> benchmark the algorithms against a deterministic nonlinear optimization
> problem. For this framework, and especially in large-scale settings, we
> understand that L-BFGS is the algorithm of choice, even though there is
> no general consensus on how to choose its step sizes. We tried to
> highlight that at the beginning of the experimental section.
>
> **Reviewer**
>
> _3\. Based on the strong and week points of this paper, I tend to reject
> it._
>
> _4\. Supporting arguments Please refer to my above arguments._
>
> _5\. Questions_
>
> _1. Can author/s please explain the first two plots in Figure 4? It seems
> to me that both Adam and RMSProp are faster reaching to the required
> gradient norm precision. The precision seems different between these two
> algorithms and the author/s claimed one. Why not use the same precision
> requirement so that we can compare these algorithms directly?_
>
> **Authors**
>
> In Figure 4 each bullet denotes the “final time” $t_f$ at which a
> precision $||g_k||<\epsilon$ was attained by two different algorithms
> for a specific task. Basically, bullets above the blue line indicate
> tasks for which the algorithm in the x-axis performs better and
> *vice-versa*. If you consider the lowest precision $\epsilon = 10^{-3}$,
> ADAM is attaining it before our algorithm. However, higher values of
> precision are not attained by ADAM and RMSprop in the course of $K$
> iterations, and for those cases we consider $t_f=\infty$. These cases
> are represented in the two boxes surrounding the main plot. These plots
> compare algorithms directly and different precision values were adopted
> to compare how the stop criteria affect the performance.
>
> **Reviewer**
>
> _2\. Can author/s provide any feedback on the third point of “weak part”?_
>
> **Authors**
>
> We hope the feedback we provided above is enough.
>
> **Reviewer**
>
> _3\. Can this proposed architecture/idea be used as a step-size policy for
> not just L-BFGS?_
>
> **Authors**
>
> Yes, and we have added some discussion in this sense to Section 3.

---

### Official Review · AnonReviewer4 · 2020-10-20
**Review on Step Size Policy Network for L-BFGS**

**Rating:** 5
**Confidence:** 4

**Review:**

** Description

This paper makes two separate contributions to machine learning: (1) It proposes a neural network to learn the step-size policy of L-BFGS and (2) it uses the algorithm for training a deep neural network.

** Pros

In practical terms it provides a solution to the problem of finding an appropriate step-size policy.  It also demonstrates that the policy out performs ADAM and RMSprop on a rather strangely chosen problem.

** Cons

Learning how to optimise by examining related problems is often a fruitless endeavour because small changes in a problem can drastically change the behaviour of optimisation algorithms.  It would have been nice to see more convincing evidence that learning a step-size polices generalises across at few more problem domains.

For me, the evaluation of their algorithm for training a neural network was slightly unconvincing.  Possibly this was just chosen as an example optimisation problem and the application shouldn't be taken too seriously (a comment to that effect would be useful).  Obviously the use of full-batch training is not that practical for most problems.  For neural network training, robustness to the noise produced by mini-batch training is important to understand.  Although ADAM and RMSProp are state-of-the-art optimisers for minibatch training when compared on full-batch it would be useful to compare with other standard approaches such as conjugate gradient, etc.  The choice of problem was puzzling.  Clearly an MLP on MNIST is not representative of modern machine learning.  It left the question of whether a small network was deliberately chosen because the algorithm does not scale to really large networks.  Again a comment about this would have strengthened the paper.

I was left with the impression that the authors were being slightly selective in their choice of problems for showing the utility of their method.  I would have liked to see more conclusive evidence in this direction and a clearer discussion of the regime where this method is likely to perform well.

** Typos

The paper is comprehensible, but would gain from being proof read by a native speaker.  This is not a consideration that affects the valuation.  Examples of rewordings that would make the paper flow slightly better are

p2 l4: good <-- well
p2 l8: exempts <-- frees
p2 l23: seek <-- search
p3 section 3 paragraph 2 line 2: it does ... <-- there does ...

---

> ### Author Response · Authors · 2020-11-23
> **Author's response to AnonReviewer4**
>
>
> **Reviewer**
>
> _\*\* Description_
>
> _This paper makes two separate contributions to machine learning: (1) It
> proposes a neural network to learn the step-size policy of L-BFGS and
> (2) it uses the algorithm for training a deep neural network._
>
> _\*\* Pros_
>
> _In practical terms it provides a solution to the problem of finding an
> appropriate step-size policy. It also demonstrates that the policy out
> performs ADAM and RMSprop on a rather strangely chosen problem._
>
> **Authors**
>
> We would like to thank the reviewer for evaluating our manuscript and
> providing useful comments.
>
> **Reviewer**
>
> _\*\* Cons_
>
> _Learning how to optimise by examining related problems is often a
> fruitless endeavour because small changes in a problem can drastically
> change the behaviour of optimisation algorithms. It would have been nice
> to see more convincing evidence that learning a step-size polices
> generalises across at few more problem domains._
>
> **Authors**
>
> Following the suggestions of this and another review, we added an
> example to investigate how the proposed strategy generalizes to other
> classes of problems. More specifically, we took the already learned
> policy (for training MLPs in MNIST) and used it for training CNNs in
> CIFAR-10. The straight generalization capacity was not evident in this
> case, as the reviewer indicated. However, after a few additional
> training steps on this other class of problems provided a policy that is
> as good as its competitors, which indicates that learning can be quickly
> transferred between problem domains.
>
> **Reviewer**
>
> _For me, the evaluation of their algorithm for training a neural network
> was slightly unconvincing. Possibly this was just chosen as an example
> optimisation problem and the application shouldn’t be taken too
> seriously (a comment to that effect would be useful). Obviously the use
> of full-batch training is not that practical for most problems. For
> neural network training, robustness to the noise produced by mini-batch
> training is important to understand. Although ADAM and RMSProp are
> state-of-the-art optimisers for minibatch training when compared on
> full-batch it would be useful to compare with other standard approaches
> such as conjugate gradient, etc. The choice of problem was puzzling.
> Clearly an MLP on MNIST is not representative of modern machine
> learning. It left the question of whether a small network was
> deliberately chosen because the algorithm does not scale to really large
> networks. Again a comment about this would have strengthened the paper._
>
> **Authors**
>
> The chosen example was inspired by one of the experiments in
> Andrychowicz et al. (2016). However, the goal of this evaluation is not
> to tackle state-of-the-art machine learning problems but rather
> benchmark the algorithms against a deterministic nonlinear optimization
> problem. For this framework, and especially in large-scale settings, we
> understand that L-BFGS is state-of-the-art even though there is no
> consensus on how to choose its step sizes.
>
>
> The idea to also use conjugate gradient within our comparisons was
> considered but, after doing some research, the preconditioning
> recommended in the book “Conjugate Gradient Algorithms in Nonconvex
> Optimization” by Radoslaw Pytlak for our setting uses an L-BFGS
> iteration at each step which, in our understanding, would result in an
> algorithm that relies on L-BFGS to perform well. Therefore, we did not
> add this comparison but added some comments regarding this point.
>
> Finally, the proposed policy scales without difficulty to large problems
> as it is $O(n+n_h)$. For instance, the problem for $(n_l,n_u)=(4, 800)$
> has $n=128,317,600$ parameters.
>
> **Reviewer**
>
> _I was left with the impression that the authors were being slightly
> selective in their choice of problems for showing the utility of their
> method. I would have liked to see more conclusive evidence in this
> direction and a clearer discussion of the regime where this method is
> likely to perform well._
>
> **Authors**
>
> Unfortunately, the theory on learned optimizers is yet scarce to the
> best of our knowledge so formal guarantees of convergence and
> performance are to be developed. However, to provide more evidence of
> the policy performance, we added a second problem where CNNs are trained
> to classify CIFAR-10 images and we drew similar conclusions to the first
> experiment. We agree with the reviewer that extensive experimentation
> would be more enlightening, and this is also to be done in future work.
>
> **Reviewer**
>
> _\*\* Typos_
>
> _The paper is comprehensible, but would gain from being proof read by a
> native speaker. This is not a consideration that affects the valuation.
> Examples of rewordings that would make the paper flow slightly better
> are_
>
> _p2 l4: good &lt;– well p2 l8: exempts &lt;– frees p2 l23: seek &lt;–
> search p3 section 3 paragraph 2 line 2: it does ... &lt;– there does
> ...\_
>
> **Authors**
>
> Thank you for putting your time into this, it was very much helpful!

---

### Official Review · AnonReviewer2 · 2020-10-28

**Rating:** 4
**Confidence:** 4

**Review:**

**Summary**:

The paper presents a novel steps-size adaptation for the L-BFGS algorithm inspired by the learning-to-learn idea. The step-size policy is determined by two linear layers which compare a higher dimensional mapping of curvature information which is trained to adapt the step-size.


**Reasons for score**:

While the idea of learning to predict a suitable step-size is intriguing and is definitely worth pursuing I am not convinced that the proposed algorithm results in an active policy that usefully adapts the step-size. There are too many concerns that I think needs to be addressed and it is not clear if the speed up improves over the reliability of a line search. I therefore vote to reject the paper in its current form.

**Pros**:

- It is clear that a lot of thought has gone into the project to come up with the policy. I think it might have merit but requires additional tests.

- The figures were first difficult to understand but once the content had been explained in the text the benefit of the chosen presentation became clear.



**Concerns**:

- My main concern is best visualized in Figure 3. Both the learned policy and the BTLS seem to mostly favour a step of 1 which raises several questions.
	1) What would be the results of using no adaptation and rely on a step of 1 (or 0.9) constantly as a baseline?
	2) The $\pi$-algorithm mostly uses a step-size of 1 which happens top be the upper boundary $\tau_M$, which means it is not clear if the policy network has learned that 1 is a good step or if it has not learned at all and the results are just due to the clipping. What would happen if $\tau_M>1$ for example?
	3) Given that both the BTLS and $\pi$ mostly use $t_k=1$, is there any intuitive explanation as to why the results between the two algorithms differ by so much in figure 3? Are there additional figures where the BTLS similarly outperforms the competition (it did reach $10^{-5}$ first in ~60% of the tasks according to table 1, column 1)? This despite the fact that BTLS is at least 1 forward pass more expensive per iteration than the policy (for a fully connected network I think that is ~50% of the iteration cost).

- The benefit of using the double network for the policy is not clear to me. What would be the result of using a single linear layer instead or a recurrent network that monitors temporal changes to the curvature information that is used as input?

- Given that the input and output dimensionality of the policy network is of low dimension it would be interesting to see what the weights and biases look like for respective policy layers. By looking at the weights it would be possible to see what curvature information makes the network decide to adjust the step-length. Does the policy learn a cosine behaviour similar to the proposed possibility in the appendix?

- Could the policy be used for another optimization algorithm for which $-g_t^\intercal d_k>0$, such as RMSprop or GD? It might be easier to understand the influence of the policy in such a setting.


Comparably minor points:

- Section 3 first paragraph ends with a statement regarding $\rho_i$ to keep the Hessian approximation s.p.d by ignoring iterations. Is this used in the implementation?
- Table 1: According to what metric is "first" defined (iteration, time, function evaluations)? It would be good to mention the average value of this metric for each optimizer at the end of $K=800$ inner steps.
- In Section 6 it says that the learning rate of Adam and RMSprop was chosen to yield fast convergence. I understand this as quickly reaching a threshold, not achieving best performance for allocated time. Is this correct? That could help explain why so many settings in table 1 and figure 4 fail to converge. Personally I think the first-order algorithms should be allowed to change the learning rate between problems to prevent them from not converging (ex. RMSprop).
- Eq.7 s.t. -> with, or is the outer problem actually constrained and I missed something?


-------------------


**Post Rebuttal**

I have considered the revised article, additional reviews and rebuttal and decided to slightly raise my score but I am still in favor of rejecting the paper. Below is a summary of my reasoning.

--------

The authors have provided a good rebuttal and I am overall pleased with the detailed response, additional experiments and figures, and overall exhibited transparency.
Unfortunately my assumption about $t_k$ seemed correct when considering the additional L-BFGS-B results, which indicate that using standard $t_k=1$ is a really strong baseline that proved difficult to beat.

I would suggest finding another set of problems where $t_k=1$ is not so good for L-BFGS or consider adapting another first-order algorithm for which it is clear that the step-length needs to change between tasks and architectures.

---

> ### Author Response · Authors · 2020-11-23
> **Author's response to AnonReviewer2, part 1**
>
> **Authors**
>
> We would like to thank the reviewer for evaluating our work, for her/his
> positive comments, and for providing constructive feedback. We hope that
> this revised version and our response help to remediate her/his
> concerns.
>
> **Reviewer**
>
> 1. _What would be the results of using no adaptation and rely on a step
> of 1 (or 0.9) constantly as a baseline?_
>
> **Authors**
>
> In our revised version, we also compare our policy with a “baseline
> L-BFGS” which adopt constant step sizes of 1. We also added more
> features to the architecture which resulted in a policy that avoids step
> sizes too close to 1 at some iterations. It seems that the new input
> vector allows the policy to become aware of update directions that may
> result in huge increases, allowing it to perform better than BTLS and
> the baseline for the problem on which it was trained.
>
> **Reviewer**
>
> 2. _The $\pi$-algorithm mostly uses a step-size of 1 which happens top be
> the upper boundary $\tau_M$, which means it is not clear if the policy
> network has learned that 1 is a good step or if it has not learned at
> all and the results are just due to the clipping. What would happen if
> $\tau_M>1$ for example?_
>
> **Authors**
>
> In the revised version, a different policy behavior is observed but, to
> investigate this point raised by the reviewer, we also tried letting it
> learn $\tau_M$. However, in our tests, a maximum step size too close to
> 1 was obtained and then we decided to fix it as, generally, $t_k=1$ is a
> good step size for this algorithm and the initial value for the BTLS.
>
> **Reviewer**
>
> 3. _Given that both the BTLS and $\pi$ mostly use $t_k=1$, is there any
> intuitive explanation as to why the results between the two algorithms
> differ by so much in figure 3? Are there additional figures where the
> BTLS similarly outperforms the competition (it did reach $10^{-1}$ first
> in $\sim$60% of the tasks according to table 1, column 1)? This despite
> the fact that BTLS is at least 1 forward pass more expensive per
> iteration than the policy (for a fully connected network I think that is
> $\sim$50% of the iteration cost)._
>
> **Authors**
>
> In the revised version the BTLS and $\pi$ are performing differently
> but, in the last version, a possible better selection of step sizes in
> early iterations could explain the cases where it performs better.
> Indeed, in the current version, the policy also prefers to be more
> cautious in the first steps what can be important to extract theoretical
> insights of how an optimal step-size policy should perform. Regarding
> the transparency, to provide a more uniform sampling of the results we
> have added 10 similar plots to Fig. 3 corresponding to the 10 first
> tasks in the test set. These figures are in Appendix B.
>
> **Reviewer**
>
> – _The benefit of using the double network for the policy is not clear to
> me. What would be the result of using a single linear layer instead or a
> recurrent network that monitors temporal changes to the curvature
> information that is used as input?_
>
> **Authors**
>
> The proposed policy is based on the idea of comparing the scalar
> projection of two vectors and then clipping the resulting value. We have
> two layers in parallel to compute two vectors and apply this operation.
> If only one layer were adopted, a second constant or learned vector
> would be necessary. However, learning a constant vector or providing an
> arbitrary one have the same results (as one can always apply a linear
> transformation to the output of the first linear layer by redefining its
> parameters). On the other hand, a second layer gives an additional
> degree of freedom to the policy with no much larger computational cost,
> as the two layers are run in parallel (in the implementation we use only
> one layer and split its output into two). Finally, we tried both cases
> (one layer + constant vector and two layers) and got better results with
> this topology.
>
>
> **Reviewer**
>
> – _Given that the input and output dimensionality of the policy network
> is of low dimension it would be interesting to see what the weights and
> biases look like for respective policy layers. By looking at the weights
> it would be possible to see what curvature information makes the network
> decide to adjust the step-length. Does the policy learn a cosine
> behaviour similar to the proposed possibility in the appendix?_

---

> > ### Author Response · Authors · 2020-11-23
> > **Author's response to AnonReviewer2, part 2**
> >
> >
> > **Authors**
> >
> > This is a good suggestion but, unfortunately, the higher-dimension of
> > the new input vector prevents us from obtaining a good graphical
> > visualization of the policy behavior. Moreover, presenting the 204
> > values defining the parameters in $\theta$ may not be very insightful.
> >
> > **Reviewer**
> >
> > – _Could the policy be used for another optimization algorithm for which
> > $-g_t^Td_k>0$, such as RMSprop or GD? It might be easier to understand
> > the influence of the policy in such a setting._
> >
> > **Authors**
> >
> > The extension of these results to the stochastic framework is a very
> > interesting future work and we stressed that in the revised version of
> > the paper. Our work, however, focuses on the study of the deterministic
> > optimization problem for which there is no good alternative to line
> > searches in the literature, to the best of our knowledge.
> >
> > **Reviewer**
> >
> > – _Section 3 first paragraph ends with a statement regarding $\rho_i$ to
> > keep the Hessian approximation s.p.d by ignoring iterations. Is this
> > used in the implementation?_
> >
> > **Authors**
> >
> > Yes, we have used this and we have made it clearer in the revised
> > version.
> >
> > **Reviewer**
> >
> > – _Table 1: According to what metric is “first” defined (iteration, time,
> > function evaluations)? It would be good to mention the average value of
> > this metric for each optimizer at the end of $K=800$ inner steps._
> >
> > **Authors**
> >
> > We have added Appendix C that gathers similar data as presented in the
> > main text but with the defined index taking into account only the final
> > value of the objective function. We drew similar conclusions for both
> > definitions.
> >
> > **Reviewer**
> >
> > – _In Section 6 it says that the learning rate of Adam and RMSprop was
> > chosen to yield fast convergence. I understand this as quickly reaching
> > a threshold, not achieving best performance for allocated time. Is this
> > correct? That could help explain why so many settings in table 1 and
> > figure 4 fail to converge. Personally I think the first-order algorithms
> > should be allowed to change the learning rate between problems to
> > prevent them from not converging (ex. RMSprop)._
> >
> > **Authors**
> >
> > The reviewer is correct but what Fig. 4 is showing is whether the
> > algorithms attained the desired precision within the pre-specified
> > horizon $K$ and, if so, how much time did it take. This does not mean
> > that $t_f=\infty$ implies in divergence, necessarily. Our interpretation
> > from this comparison is that RMSprop and Adam, even when tuned for
> > achieving the fastest convergence, fail to overcome the competitors in
> > this setting in terms of reaching gradient-base stopping criteria first.
> > We do agree that these algorithms should be allowed to change their
> > learning rates between different problems to make the best out of them
> > but that would not be a general tunning and therefore, not comparable
> > with the proposed policy which is not trained task-wise.
> >
> > **Reviewer**
> >
> > – _Eq.7 s.t. -&gt; with, or is the outer problem actually constrained and
> > I missed something?_
> >
> > **Authors**
> >
> > In this case, we have adopted a formal representation of the
> > optimization problem where the iterates could be regarded as decision
> > variables (as they depend on the policy parameters). However, in
> > practice, the “s.t.” can be regarded as a “with” for implementation
> > purposes.

---

### Official Review · AnonReviewer3 · 2020-10-29
**Learn to learn approach for L-BFGS step size parametrization**

**Rating:** 5
**Confidence:** 3

**Review:**

The paper studies a problem of learning step-size policy for L-BFGS algorithm. This paper falls into a general category of meta-learning algorithms that try to derive a data-driven approach to learn one of the parameters of the learning algorithm. In this case, it is the learning rate of L-BFGS. The paper is very similar in nature to the papers of Ravi & Larochelle, MAML and Andrychowicz.

My biggest issue with this paper is the evaluation. The paper itself cites in the introduction:  "...for large-dimensional problems this procedure is likely to become the bottle-neck of the optimization task". However, the paper doesn't not provide necessary evaluation to help resolving this issue. In fact, it is not very clear at all that the proposed method would work on a more general scenario when the evaluation dataset is wildly different from training the dataset.

I'm also a bit surprised to see this paper tackling specifically L-BFGS algorithm, instead of more general case of learning rate parameter for any gradient basis algorithm. I would be curious to learn what is so special about L-BFGS that made the authors chose it. After all, the paper (and L-BFGS) deals with deterministic optimization problems on a full batch which limits the applicability of the paper.

---

> ### Author Response · Authors · 2020-11-23
> **Author's response to AnonReviewer3**
>
>
> **Reviewer** _The paper studies a problem of learning step-size policy for L-BFGS
> algorithm. This paper falls into a general category of meta-learning
> algorithms that try to derive a data-driven approach to learn one of the
> parameters of the learning algorithm. In this case, it is the learning
> rate of L-BFGS. The paper is very similar in nature to the papers of
> Ravi & Larochelle, MAML and Andrychowicz._
>
> _My biggest issue with this paper is the evaluation. The paper itself
> cites in the introduction: “...for large-dimensional problems this
> procedure is likely to become the bottle-neck of the optimization task”.
> However, the paper doesn’t not provide necessary evaluation to help
> resolving this issue. In fact, it is not very clear at all that the
> proposed method would work on a more general scenario when the
> evaluation dataset is wildly different from training the dataset._
>
> **Authors**
> First, we would like to thank the reviewer for her/his assessment of our
> work and thoughtful comments. In the revised version of our manuscript,
> we added a second experiment where we learned our policy by training MLP
> classifiers for MNIST images and, subsequently we applied the policy to
> a different problem which is training a CNN for CIFAR-10 images. Indeed
> a direct generalization of our policy to this different problem is not
> assured but, after a few additional training steps on other classes of
> problems, the policy can quickly learn how to perform well on these
> problems. This shows that in our framework, although learning cannot be
> generalized from one class to another, it may be efficiently
> transferred.
>
> The correspondent part is:
>
> >“ Finally, as a last experiment, we applied the learned policy and these
> competitors to a class of tasks comprising the training of a
> Convolutional Neural Network (CNN) to classify images in CIFAR-10, see
> (Krizhevsky et al., 2009). The adopted architecture is described in
> Zhang (2016) but sigmoid activation functions were replaced by ReLU to
> make this problem even more distant from the one $\pi$ was trained on. A
> training and a test set of tasks, $\mathcal{T}\_{train}^C $ and
> $\mathcal{T}\_{test}^C$, were built similarly to $\mathcal{T}\_{train}$
> and $\mathcal{T}\_{test}$ but using images in CIFAR-10 instead.
> Evaluating these algorithms in $\mathcal{T}\_{test}^C$ and computing the
> index $\mathcal{I}\_a(f)$ for each task allows us to present the first
> box plot in Fig. 6. This figure indicates that $\pi$ do not perform as
> good as before in these problems. This could be expected as a different
> architecture directly affects the nature of the objective function. To
> investigate whether the learned policy $\pi$ can be used as a warm-start
> for the training a new policy $\pi^C$, we perform additional training
> steps on $\pi$ corresponding to 10 epochs in the training set
> $\mathcal{T}\_{train}^C$, but eliminating 5/6 of its tasks. This is done
> to show that even with very low effort placed in this retraining phase
> and considering fewer learning data, we can benefit from previous
> learning to speed-up new training procedures. The corresponding results
> are presented in the second box plot of Fig. 6, which shows that the new
> policy $\pi^C$ performs comparably to the competitors. Certainly,
> further investigation is required but this suggests that some learning
> can be transferred across distinct problem domains.”
>
>
> **Reviewer**
> _I’m also a bit surprised to see this paper tackling specifically L-BFGS
> algorithm, instead of more general case of learning rate parameter for
> any gradient basis algorithm. I would be curious to learn what is so
> special about L-BFGS that made the authors chose it. After all, the
> paper (and L-BFGS) deals with deterministic optimization problems on a
> full batch which limits the applicability of the paper._
>
>
> **Authors**
> We chose L-BFGS because as it is often the algorithm of choice in
> deterministic unconstrained optimization. The examples we present may
> not be state-of-the-art machine learning problems, which rely more often
> upon stochastic optimization approaches but the choice is to point the
> gap in the literature regarding the selection of a step-sizes for this
> algorithm to perform as good and as automatic as possible.

---

### Decision · Program_Chairs · 2021-01-07
**Final Decision**

**Decision:**

Reject

**Comment:**

The paper presents a novel procedure to set the steps-size for the L-BFGS algorithm using a neural network.
Overall, the reviewers found the paper interesting and the main idea well-thought. However, a baseline that was proposed by one of the reviewers seems to be basically on par with the performance of the proposed algorithm, at least in the experiments of the paper. For this reason, it is difficult to understand if the new procedure has merit or not. Also, the reviewers would have liked to see the same approach applied to different optimization algorithms.

For the reasons, the paper cannot be accepted in the current form. Yet, the idea might have potential, so I encourage the authors to take into account the reviewers' comments and resubmit the paper to another venue.